# Multispectral Images for Drought Stress Evaluation of Arabica Coffee Genotypes Under Different Irrigation Regimes

**DOI:** 10.3390/s24227271

**Published:** 2024-11-14

**Authors:** Patrícia Carvalho da Silva, Walter Quadros Ribeiro Junior, Maria Lucrecia Gerosa Ramos, Maurício Ferreira Lopes, Charles Cardoso Santana, Raphael Augusto das Chagas Noqueli Casari, Lemerson de Oliveira Brasileiro, Adriano Delly Veiga, Omar Cruz Rocha, Juaci Vitória Malaquias, Nara Oliveira Silva Souza, Henrique Llacer Roig

**Affiliations:** 1Faculdade de Agronomia e Medicina Veterinária, Universidade de Brasília, Brasília 70910970, Brazil; 2Embrapa Cerrados, Empresa Brasileira de Pesquisa Agropecuária, Planaltina 73310970, Brazil; lbrasileiro@ua.pt (L.d.O.B.);; 3Instituto Tecnológico de Agropecuária de Pitangui (ITAP), Empresa de Pesquisa Agropecuária de Minas Gerais, Pitangui 35650000, Brazil; 4Laboratório de Geoprocessamento,, Instituto de Geociências, Universidade de Brasília, Brasília 70910970, Brazil; 5Embrapa Café, Empresa Brasileira de Pesquisa Agropecuária, BR 020, Km 18, Brasília 73310970, Brazil

**Keywords:** water supply, *Coffea arabica*, UAV, Cerrado

## Abstract

The advancement of digital agriculture combined with computational tools and Unmanned Aerial Vehicles (UAVs) has opened the way to large-scale data collection for the calculation of vegetation indices (VIs). These vegetation indexes (VIs) are useful for agricultural monitoring, as they highlight the inherent characteristics of vegetation and optimize the spatial and temporal evaluation of different crops. The experiment tested three coffee genotypes (Catuaí 62, E237 and Iapar 59) under five water regimes: (1) FI 100 (year-round irrigation with 100% replacement of evapotranspiration), (2) FI 50 (year-round irrigation with 50% evapotranspiration replacement), (3) WD 100 (no irrigation from June to September (dry season) and, thereafter, 100% evapotranspiration replacement), (4) WD 50 (no irrigation from June to September (water stress) and, thereafter, 50% evapotranspiration replacement) and (5) rainfed (no irrigation during the year). The irrigated treatments were watered with irrigation and precipitation. Most indices were highest in response to full irrigation (FI 100). The values of the NDVI ranged from 0.87 to 0.58 and the SAVI from 0.65 to 0.38, and the values of these indices were lowest for genotype E237 in the rainfed areas. The indices NDVI, OSAVI, MCARI, NDRE and GDVI were positively correlated very strongly with photosynthesis (A) and strongly with transpiration (E) of the coffee trees. On the other hand, temperature-based indices, such as canopy temperature and the TCARI index correlated negatively with A, E and stomatal conductance (gs). Under full irrigation, the tested genotypes did not differ between the years of evaluation. Overall, the index values of Iapar 59 exceeded those of the other genotypes. The use of VIs to evaluate coffee tree performance under different water managements proved efficient in discriminating the best genotypes and optimal water conditions for each genotype. Given the economic importance of coffee as a crop and its susceptibility to extreme events such as drought, this study provides insights that facilitate the optimization of productivity and resilience of plantations under variable climatic conditions.

## 1. Introduction

Coffee, the second most traded commodity in the world, plays an important economic role in several Latin American, African and Asian countries [1]. With 38.1% of the global production, Brazil was the world’s largest producer and exporter of coffee beans in the 2023/2024 growing season [2]. The total acreage for coffee in the country (*Coffea arabica* and *canephora*) was 2.25 million hectares in 2023 [3]. This amount shows the high economic and social relevance of Brazilian coffee farming, which is a source of job generation and income increase [4].

Coffee is a perennial crop that has been produced for several years, with harvests alternating between higher and lower productivity [5,6]. Planting is done with seedlings, and fruiting begins 2 to 4 years after planting [6]. Harvests occur between May and September, depending on the region and the fruit maturation cycle [7].

The Cerrado region has a long dry period that can affect coffee development. Extreme weather events, e.g., prolonged droughts and frosts, have affected production, and even 60% of coffee species are threatened with extinction [5,7].

In Brazil and other countries, multispectral images obtained by UAVs are used in the coffee tree for several purposes, such as discriminating coffee plants with unripe fruits from ripe fruits [8,9] and predicting plant water stress using wavebands in the VIS/NIR region [10]. In addition, multispectral images associated with morphological characteristics of grains are also used to discriminate coffee species [11].

This information obtained by sensors is useful not only in phenotyping carried out in the selection of genotypes in breeding programs under water stress but also in decisions in the area of plant management and irrigation. It is, therefore, important to monitor coffee plantations to minimize production losses by means of tailored management practices [4].

To monitor the water levels on coffee plantations, techniques must be available that allow the assessment of large areas or regions [4]. The most commonly used method to decide to begin the coffee harvest is based on visual inspection, which is time-consuming, laborious and not representative of the entire cultivated area.

In this context, technology and tools based on remote sensing principles to obtain this information have been further developed, and data can now be acquired through various media such as satellites, drones or other aircraft [12]. The spatial, spectral, temporal and radiometric image resolution of the sensors, such as multispectral and hyperspectral sensors used in UAVs, are high and, combined with advanced computer programs, is extremely helpful to infer a range of spatio-temporal information for any location on the planet [13].

Although satellite sensors can retrieve data inexpensively and have proven effective for large-scale crop phenotyping [14,15,16], their spatial and temporal resolution is rather limited for the analyses [17]. On the other hand, sensors mounted on unmanned aerial vehicles (UAVs) are more efficient. Multispectral sensors attached to UAVs are economically viable for plot-level phenotyping since the high spatial, spectral and temporal resolution data are accurate enough for crop breeding [18,19,20].

The use of the most diverse vegetation indices by coffee growers has proved promising in the assessment of various aspects of crops, including physical characteristics and crop health, by monitoring pest and disease incidence [21]. There are also vegetation indices that correlate well with the biophysical vegetation parameters and are widely used to estimate biomass and changes in crop growth and development [22,23]. Martins et al. [24] developed a vegetation index (VI) to monitor coffee ripeness using aerial images. After image acquisition, the coffee ripeness index and five other VIs were computed. These VIs, therefore, indicate biotic and abiotic stress [10,25]. The Normalized Difference Vegetation Index (NDVI), for example, is widely used, as it indicates the general state of vegetation [26,27]. It is an index used as a phenological indicator of crop development over months. It can be used as an indicator of drought because it is correlated with the green biomass of above-ground vegetation [28].

Leaf reflectance can be used as an indicator of plant function because the reflected light depends directly on the pigment composition of the leaf (chlorophylls and xanthophyll), which may reflect the physiological state of plants [29]. Typically, vegetation indices are derived from reflectance bands based on RGB (red, green and blue), infrared and other narrow wavelength spectra. These specific reflectances are commonly measured by remote sensing with visible, multispectral and hyperspectral cameras [29,30,31,32].

Recent studies demonstrated the usefulness of hyperspectral remote sensing indices in the evaluation of biophysical variables of vegetation in agriculture [33]. For the authors, indices such as the TCARI and the OSAVI represent the combined response to variations in several environmental and vegetation properties, such as the Leaf Area Index (LAI), leaf chlorophyll content, canopy shadows, and soil reflectance.

There are several VIs used to assess terrestrial ecosystems. However, there are limitations in applying these indices to dryland research due to their low sensitivity in low vegetation cover sites [34]. This author developed a new vegetation index, the Generalized Difference Vegetation Index (GDVI), to examine its applicability to dryland environment assessments and concluded that the GDVI has great potential in site characterization as well as land assessments in drought-affected environments. Dash and Curran [35] concluded that the MTCI is a suitable index for estimating chlorophyll content. However, they suggest further research to investigate the value of the MTCI with data from controlled and field experiments to evaluate the performance of the index on a larger scale.

Several authors have studied possible applications and the potential of vegetation indices for the evaluation of coffee [36,37,38,39,40]. This study evaluated and validated multispectral images to determine vegetation indices for the analysis of drought stress and tolerance of Arabica coffee genotypes in response to different irrigation levels in the Brazilian Cerrado.

## 2. Materials and Methods

### 2.1. Study Area and Experimental Design

This study was carried out on a coffee plantation in an experimental area of EMBRAPA Cerrados-Planaltina-Distrito Federal (15°35′ S, 47°42′ W; at 1007 m asl) (Figure 1). According to Köppen’s classification, the regional climate is Aw, [41], with two well-defined seasons (dry and rainy). Annual averages are 21.1 °C and 1345 mm of precipitation (Figure 2). The soil of the experimental area was classified as clayey Oxisol [42], with a soft undulating relief and a clayey texture.

The experimental area consisted of a 0.74-ha plantation of the species *Coffea arabica* L. (Figure 1). The crop was planted in 2015 at a row spacing of 3.50 m and plants spaced 0.50 m apart. Three genotypes were chosen, and the same genotypes were used by Silva et al. [43] in order to use the physiology data used by these authors. The genotypes were Catuaí 62, E237 and Iapar 59. Iapar 59 is considered slightly tolerant to water stress [44]. E237 was chosen because it is from Ethiopia and is adapted to the region of origin of the coffee species, while Catuai 62 is adapted to irrigated conditions in the Brazilian Cerrado region [45]. Seven water regimes described below were tested, although some were further explored in the results: (1) Rainfed (no irrigation during the year). (2) WD1 50% (irrigation with the suspension of irrigation from April to September and after water stress, the replacement of 50% of evapotranspiration). (3) WD2 50% (irrigation with the suspension of irrigation from June to September and after water stress, the replacement of 50% of evapotranspiration). (4) FI 50% (year-round irrigation with a 50% replacement of evapotranspiration). (5) FI 100% (year-round irrigation with a 100% replacement of evapotranspiration). (6) WD2 100% (irrigation with the suspension of irrigation from June to September and after water stress, the replacement of 100% evapotranspiration. (7) WD1, 100% (irrigation with the suspension of irrigation from April to September and after water stress, the replacement of 100% evapotranspiration).

The irrigated treatments were watered with irrigation and precipitation (Figure 1). In the irrigated area, one water regime was used for each treatment described below. In Figure 1, we show seven water regimes and eight genotypes, but, for this paper, only the three genotypes and the five water regimes (treatments 1, 3, 4, 5, 6) mentioned below were taken into consideration. Precipitation was greater in 2020 in January, February, October and November, and, in the dry period, the temperature was slightly lower in 2020 (Figure 2).

For the irrigated treatments, the coffee trees were watered with mechanized Mobile Lateral Line sprinkler irrigation. The criterion for the irrigation level to be applied was based on the climatological water balance and the Cerrado Irrigation Monitoring Program [46], from which data on soil water availability in the experimental area and crop coefficients for coffee defined by Guerra et al. [47] were extracted. This information provided the amount of water to be applied and the interval between irrigation applications for adequate management. The Penman–Monteith method [48] by the aforementioned Program was used to estimate reference evapotranspiration (Eto).

### 2.2. Sensors and Image Acquisition

In August 2019, aerial photographs of the experimental area were taken with RedEdge-MX^®^ cameras (Table 1) for multispectral imaging and FLIR Duo Pro^®^ for thermal imaging (Table 2). In August 2020, Parrot Sequoia^®^ cameras (Table 3) were used for multispectral imaging and FLIR Duo Pro^®^ for thermal imaging. All cameras were mounted on a DJI M600 hexacopter (SZ DJI Technology Co., Shenzhen, China). The images were captured during the peak water stress period.

Images with 80% longitudinal and lateral overlaps were collected, resulting in a spatial resolution of 0.04 m pixel^−1^. A flight plan was created in autopilot mode with the DroneDeploy^®^ (San Francisco, CA, USA) application. The UAV was set to photograph the experimental area at a height of 60 m, 0.04 m pixel^−1^ and 3 m s^−1^. The images were taken around noon to reduce shadow effects, when solar radiation on the Earth surface is more homogenous. After the flight, the raw images were stored in a flash drive and later transferred to a computer in TIF format for pre-processing.

The cameras used have different specifications that can influence the spectral data obtained, mainly in relation to the wavelengths captured and also in relation to the sensitivity of the sensors, but we applied radiometric corrections to the images to ensure comparability, taking into account the lighting conditions and sensor configurations. For these corrections, we used parameters of the cameras, irradiance data and calibration panels placed on the ground during flights, which helped to normalize the data and reduce discrepancies caused by differences between spectral bands and camera resolutions, so pre-processing methods were applied to minimize the impact of these variations on the spectral data.

### 2.3. Pre-Processing of the Images

The photos stored on the computer were loaded into the PIX4Dmapper software, version 4. and orthophotos were generated from the set of raw and georeferenced images to represent the entire experimental area (Figure 1). Orthomosaics corresponding to reflectance (multispectral) and temperature (thermal) values were computationally generated using ‘Structure-from-Motion’ (SfM) photogrammetry. This tool automatically identifies similar features in a series of overlapping images by a bundle adjustment procedure [49].

To improve the data quality of the multispectral sensor, radiometric corrections were applied, based on camera parameters, irradiance data and corrections with a calibration panel, which was placed on the ground of the experimental area at the time of the flight. Subsequently, the PIX4Dmapper pre-processing software used the photo bands of the calibration panel captured prior to each flight, and the irradiance information collected by the solar sensor was used to normalize the data during the flight. This step allowed a comparison of images under different lighting conditions at the time of the flight.

Based on overlapping images, PIX4Dmapper calculated reflectance values for each pixel of the orthomosaic, based on a weighted average of pixels from all original images that correspond to that specific pixel, assigning greater weight to images where the pixel is more central. The value of each image pixel depends on several factors, including sensor settings, sensor properties, and scene conditions [50].

Each plot was marked and separated in both orthophotos using QGis^®^ software Version 3.8 Zanzibar, resulting in images and values representing a single experimental plot. The different bands generated images of the experimental area. In addition, the pixel count, sum, mean, median, minimum and maximum values were determined with the Zonal Statistic tool available in QGis^®^, as well as the variance of the different VIs.

### 2.4. Vegetation Indices, Leaf Gas Exchange and Productivity

In this study, eight vegetation indices were used, all calculated by the QGIS raster calculator (equations in Table 4).

The productivity of each coffee genotype and Pearson’s correlation between vegetation indices and leaf gas exchange were computed. Productivity and leaf gas exchange data were taken from Silva et al. [43].

### 2.5. Statistical Analysis

In this study, we aimed to evaluate the impact of water regimes and coffee genotypes on vegetation index. For that aim, the experiment consisted of five water treatments: for each water regime, three genotypes were arranged in randomized blocks with four replications. Initially, an analysis of variance (ANOVA) was performed in randomized blocks for each coffee genotype and water regime. Subsequently, the Hartley F Max test was applied, using the mean square error of each variable in each treatment. The result of the Hartley F Max test was used to validate the experiments as a group, and the combined analysis of groups of experiments was used. After that, water regimes and genotypes were considered as sources of variation, and the ANOVA was rerun.

Additionally, a residual analysis was performed to check for the presence of outliers, and the assumptions of normality and homogeneity of variance were tested. The Shapiro–Wilk test was used to verify normality, and the Bartlett test was employed to assess the homogeneity of variance. An analysis of variance was performed using the F and Tukey’s tests (*p* < 0.05) to compare the means. For statistical analysis, software RStudio, version 4.3.1, was used, and the graphs were created using Sigmaplot software, version 10.

## 3. Results

### 3.1. Performance of Vegetation Indices for Each Coffee Genotype Under Different Water Regimes

The performances of the vegetation indexes (Vis) differed between the two years, possibly due to the variation in environmental factors in 2019 and 2020, e.g., in temperature and rainfall, which affected the response of the genotypes to different water regimes (Figure 3, Figure 4, Figure 5 and Figure 6). The indices NDVI, SAVI, MCARI, GDVI and MTCI were higher in the full irrigation treatments (FI 100 and FI 50), with a significant reduction under WS 100 and 50 and the rainfed treatments in 2019 (Figure 3A,B,E,H,G).

In the same year, the NDRE and GNDVI were highest in the FI 50 treatment, whereas under FI 100, only Iapar 59 performed better than in water stress treatments (Figure 3C,F). The results for index TCARI were similar in both years. In 2020, the TCARI was highest under water stress, whereas in 2019, it was lowest under full irrigation (FI 100 and FI 50), except for genotype Iapar 59. The indices NDVI, SAVI, NDRE and MCTI were lowest in 2020 (Figure 4A,B,C,G). The NDVI, SAVI, MCARI, GDVI were highest under FI50 (Figure 4A,B,E,H). Evaluating the indices NDVI, SAVI, NDRE, MTCI and GNDVI for the genotypes within each treatment, genotype E237 had the lowest performance under water stress. Under full irrigation (FI100 and FI50), the tested genotypes did not differ between the years of evaluation. In general, the index values of Iapar 59 exceeded those of the other genotypes.

The thematic maps of VIs in the evaluated period represented both the variability of the different genotypes and the spectral responses of the indices in 2019 and 2020 (Figure 5 and Figure 6). As the evaluations were performed in the period of most severe water stress, none of the VIs was saturated, which allowed a clearer discrimination. For the images of 2019, the OSAVI, TCARI and NDVI differentiated the full irrigation regimes (FI 100 and FI 50) better than the water stress treatments (Figure 5).

Differences in indices between years, indicated in Figure 5 and Figure 6, are due to changes in the environment (temperature and precipitation).

### 3.2. Correlation Between Vegetation Indices and Physiological Variables

Pearson correlation measures the strength of the linear relationship between variables and not the overall relationship (linear and nonlinear). Vegetation productivity has been shown to vary nonlinearly with some vegetation and climate indices (Figure 7 and Figure 8).

A correlation analysis was performed to determine the indices that correlated best with the physiological variables. The indices NDVI (0.87), OSAVI (0.90), MCARI (0.92) and GDVI (0.90) had a positive, very strong and highly significant correlation with photosynthesis (A) in 2019 (Figure 7). Similarly, the same indices were strongly correlated with plant transpiration (E). On the other hand, canopy temperature (T) and the TCARI were negatively correlated with A, gs and E. The TCARI was only positively related to temperature (0.72). The MTCI had no significant correlation with physiological variables and productivity but only a moderate correlation with the NDRE and GNDVI. In general, the indices were weakly correlated with productivity and stomatal conductance (Figure 7).

In 2020, probably due to changes in environmental conditions, the correlation coefficients were lower. Between the indices NDVI, OSAVI, MCARI and GDVI and A, E and gs, only moderate correlations were found (Figure 8).

### 3.3. Principal Component Analysis

An exploratory analysis of principal components was carried out to find relationships between vegetation indices and coffee tree physiology and productivity for different water regimes and genotypes (Figure 9A–D). The behavior of physiological variables and VIs were similar for the different genotypes (Figure 9B,D). With regard to the water regimes, the treatments with year-long irrigation (FI50 and FI100) formed differentiated groups for most of the VIs and physiological variables evaluated (Figure 9A,C).

For the NDVI, OSAVI, SAVI, MCARI, Yield, gs, E and A, converging vectors were found in FI 100 and FI 50. Except for gs, the NDVI, OSAVI, SAVI, MCARI, Yield, E and A had similar Dim.1 values, indicating that they influenced each other positively. On the other hand, in the water stress and rainfed treatments, TCARI and T were the most strongly related indices to water stress regimes (Figure 9A,C). The values of the TCARI and T were close to those of A, the NDVI, the SAVI, the GDVI and the OSAVI. However, they are on opposite sides, which indicates a negative relationship between them. The MCARI was most positively related to productivity, gs, E and A, while the TCARI and T were negatively associated (Figure 9A,C).

## 4. Discussion

### 4.1. Vegetation Indices to Identify Drought Stress in Coffee Trees

Eight indices, widely applied in agriculture, were included in this study. The maps shown in this study provide visual information that is readily available, mainly for the NDVI, OSAVI and GNDVI, for 2019 and 2020 (Figure 5 and Figure 6). However, it is necessary to obtain the value of the indexes and make a statistical analysis to eliminate analytical bias. Thus, large-scale statistical analyses were used in this study.

Silva et al. [55], using the results of this experiment, concluded that coffee cultivation is economically viable for the Brazilian Cerrado, as long as irrigation is used to complement natural precipitation, even in years of low prices.

Exposure to drought stress negatively affects coffee growth and productivity. The severity of drought-induced damage determines plant recovery kinetics [56]. In the current scenario of recurrent drought events, recovery kinetics are key in predicting the stress tolerance potential and survival status of a given plant cultivar [56]. According to Ramasamy et al. [57], in response to water stress in resistant or tolerant plants, drought-defense strategies are triggered, termed morphological and physicochemical/biochemical mechanisms. In this study, evaluations took place shortly before coffee harvest, a time when the trees are exposed to a high level of drought stress under rainfed or water stress conditions. At this stage, the morphological and physiological performance of plants can generally be measured by eight vegetation indexes (NDVI, MCARI, TCARI, NDRE, GNDVI, GDVI and MTCI). Several of these indexes are related to plant development and physiology. The NDVI is one of the most used to study plant development and yield.

In a study to monitor the vegetative state of coffee in the harvest phase based on VIs, Chedid et al. [38] concluded lower degrees of variability and a better fit for coffee of the indices NDVI and SAVI. However, the NDVI is sensitive to effects of soil brightness, soil color, atmosphere, the presence of clouds and leaf canopy shade, which requires calibration [58].

The NDVI is the most widely used index in agriculture for decision making. In a study to monitor the vegetative state of coffee in the harvest phase based on VIs, Chedid et al. [38] concluded lower degrees of variability and a better fit for coffee of the indices NDVI and SAVI. The NDVI values ranged from 0.87 to 0.58 and SAVI from 0.65 to 0.38 and were lowest in rainfed areas with genotype E237 (Figure 4 and Figure 5). Chedid et al. [38] found NDVI values of 0.2 to 0.8 and SAVI of 0.32 to 1.18 in coffee grown in Cerrado areas in Minas Gerais. In this study, the use of the NDVI to indicate vegetation stress and soil moisture has already been reported elsewhere [26,59,60], as the NDVI is sensitive to chlorophylls and other plant pigments responsible for absorbing red band radiation [61]. Lower NDVI indices under water stress indicate lower chlorophyll leaf contents and have been used to assess drought status since the 1970s [4], when this index was proposed by Rouse et al. [28]. In addition, Mbatha and Xulu [59] also demonstrated the applicability of the NDVI to monitor the impact of intense drought caused by El Niño in South Africa.

For coffee, Barata et al. [39] observed that the NDVI, MCARI1 and GNDVI decreased during drought and increased in the rainy season, with a decline as of May and lowest values in September. Thus, the measurement of VIs in the spring or summer months could result in misleading interpretations about the performance of coffee genotypes in response to the different water regimes studied.

To evaluate the physiological state of plants under water regimes in studies, the method currently used is the measurement of gas exchange with a portable open-flow gas exchange system (IRGA—LI-6400XT; LI-COR Inc., Lincoln, NE, USA). However, this method is rather limited because it is time-consuming, observation-based and user-dependent, which restricts the monitoring of large areas. Therefore, vegetation indices were used in this study to identify sustainable irrigation management that would optimize water-use efficiency and indicate more drought-tolerant genotypes. It is worth mentioning that the advancement of digital agriculture combined with computational tools and UAVs in coffee allows data collection to establish reliable indices and biophysical parameters [24,62].

Due to the biennial bearing of Arabica coffee, the VIs computed at a specific time are inversely related to the yield of that year and of the following (or earlier) [37,40]. In addition, according to these authors, this relationship is reversed after the onset of rainfall, when VIs change abruptly and their correlation with yield is inverted, becoming negative. Thus, one of the possible causes of variability observed in the field is the existence of trees with different patterns of yield potential due to their unequal bienniality (Figure 4 and Figure 5).

When coffee trees respond to drought stress as in this study, the Normalized Difference Vegetation Index (NDVI) values tend to decrease as the drier conditions alter the biophysical conditions of the leaves. Although the near-infrared and red bands are not directly correlated with water content, they are linked to chlorophyll and other biophysical parameters such as above-ground net primary production, green leaf biomass and plant photosynthetic activity [63], and these variables are related to drought stress [64].

Apart from the NDVI, other ways of understanding crop performance through remote sensing are used, for example, to detect water stress in crops by indices based on the remote sensing of near- and mid-infrared wavelength ranges [63]. Although the authors concluded that these indices cannot be used to detect water stress remotely, from the time of their study to this moment, the technology for collecting, processing and analyzing data has evolved greatly. In this study, the technique of multivariate analysis was included to quantify several leaf components simultaneously. These variables are linked to drought stress since, during water stress, crops make adjustments in functional, structural and biochemical characteristics [65].

However, there is also difficulty in monitoring drought stress by VIs, since this response is observed when notable damage to the crop has already occurred [27]. To overcome the problem, an interesting initiative would be to study these indices throughout the entire plant production cycle to identify the harm caused by a lack of water in early cultivation stages. Due to the different climatic conditions throughout Brazil and the different conditions of coffee trees, e.g., the development stage and even the cultivar, there can be great variation in the values of the crop coefficients (Kc), both spatially and temporally [62]. According to the authors, it is possible to estimate the biophysical data of coffee generated remotely through UAVs.

Under full irrigation (FI 100), most indices achieved the highest values, reflecting the importance of adequate water supply for the growth and development of coffee trees (Figure 3, Figure 4 and Figure 5). On the other hand, the water stress (WS 100 and WS 50) and rainfed water regimes had a negative impact on the performance of coffee cultivation in 2019, which was aggravated in 2020.

Bernardes et al. [66] monitored the effect of bienniality on coffee production using remote sensing images (MODIS). Coffee productivity and leaf biomass were negatively correlated, as shown by the time series of vegetation indices derived from satellite images. The authors evaluated the indices EVI and NDVI from 2002 to 2009 in the south of Minas Gerais. Correlations were observed between the variation in coffee productivity per plot and the variation in VIs for pixels overlapping the same coffee plots. Although the correlations were not sufficient to estimate coffee productivity based exclusively on VIs, the trends observed in this study adequately reflected the biennial effect on coffee productivity. Barata et al. [39] drew attention to the influence of climate change on coffee production and how studies with UAVs can be fundamental in this new production context. For the authors, the use of artificial intelligence, machine learning and research with UAVs are options that could estimate fruit volume and maturation, harvest prediction, and identification of damage caused by extreme climatic events and the creation of adaptation strategies to climate change. The data presented in this study indicate that agricultural practices that make coffee trees more resilient and adaptable, capable of optimizing the use of water resources and maximizing productivity despite climate change and environmental variations, are crucial for the success of cultivation. Choosing the correct genotype and appropriate irrigation strategy can, therefore, have a significant impact on the sustainability and effectiveness of coffee production.

When coffee trees do not receive enough water, fewer flowers and, consequently, fewer fruits may be produced, resulting in yield reduction [67]. Furthermore, the fruit size is smaller and their quality lower, resulting in reduced coffee acidity, aroma and body, which affects the tasting experience. All these consequences reduce the market value and profitability for coffee producers as well as the entire production chain.

In this study, results differed considerably between 2019 and 2020 (Figure 7 and Figure 8). As mentioned previously, environmental differences contributed to this outcome. Despite the numerical differences, the results and direction of the correlation curves between the studied indices and variables were very similar in 2019 and 2020, indicating that there was no major environmental interaction that would have influenced the interpretation of the correlations between the different variables and coffee productivity.

The variables photosynthesis, stomatal conductance, transpiration and canopy temperature were correlated with each other to different degrees, which shows the interconnected processes of plant physiology in 2019 (Figure 7). Due to the direct participation of physiological variables in the processes of gas and water exchange in plant photosynthesis, a complex interaction between plant physiology and the spectral responses detected by the images can be inferred. Canopy temperature was negatively correlated with most vegetation indices, suggesting negative impacts of higher leaf temperatures on transpiration, photosynthetic activity and plant performance as a whole. Bento et al. [37] stated that VIs provide information about vegetative growth, along with variations in air temperature and atmospheric precipitation. However, those authors observed no trend of variation similar to the VIs in chlorophyll values, in contrast to our observations. Vegetation indices (VIs) are commonly used in agricultural monitoring due to their power to highlight the intrinsic characteristics of vegetation, which are related to the reflection of green, which indicates the vigor level of plants [24]. According to Fabri et al. [68], of the tested VIs, the biophysical variables of coffee, such as height and diameter, are most strongly correlated by the NDVI. In this study, the NDVI and the physiological variables (photosynthesis and transpiration) were also strongly correlated in 2019. This finding can be explained by the plants with greater vegetative vigor and better photosynthetic potential. Thus, the most irrigated plants, without drought stress and with more biomass, had higher NDVIs (Figure 6).

Chedid et al. [38] observed that the NDVI had the best correlation with productivity in the dormancy and flowering phases in both study years. This index explained between 62% and 89% of the variation in productivity data observed in the dormancy and 58% to 73% in the flowering phase. However, the authors observed a lower relationship with the productivity of the SAVI than the NDVI in the 2013/2014 and 2014/2015 growing seasons. In this study, both the SAVI and NDVI had a median correlation with productivity (Figure 7 and Figure 8).

In 2019, the correlation between productivity and indices such as the NDVI (0.47), OSAVI (0.47), MCARI (0.34), NDRE (0.50), GNDVI (0.41) and GDVI (0.45) were positive and moderate. This suggests that these indices can be useful indicators of the potential yield of coffee trees (Figure 7). The correlation between productivity and these indices was as expected, since all these indices measure similar aspects of coffee tree vigor. For the NDRE, correlations with all studied variables were high, which shows the relevance of evaluating the chlorophyll content of plants. Bento et al. [37] also reported good results with this index. The authors emphasized that the NDRE explained the phenomena of water stress effects on coffee, especially in relation to water stress and temperature, which hampers the leaf tissue directly. The OSAVI can eliminate the influence of the soil surface and is usually used to calculate biomass, leaf nitrogen content and chlorophyll content [58]. Thus, productivity correlations with these indices are important to analyze the performance of coffee trees.

The variable canopy temperature was negatively correlated with photosynthesis (−0.63), stomatal conductance (−0.62) and transpiration (−0.62). The negative corretation between canopy temperature and gas exchange suggests that higher temperatures may be associated with the reduced efficiency of metabolic processes or with thermal stress that triggers stomatal closure in coffee trees. Canopy temperature was also negatively correlated with the NDVI (−0.63), OSAVI (−0.64), MCARI (−0.61), GDVI (−0.58) and productivity (−0.62). In other words, high temperatures may be negatively associated with plant performance and chlorophyll content, possibly due to water and thermal stress. This finding reinforces the importance of monitoring environmental conditions, especially in relation to water supply, to optimize the productivity of coffee plants.

Among these reflectance indices, the MCARI is correlated with photosynthetic parameters, as the correlation and principal components show (Figure 7, Figure 8 and Figure 9). It can be used as a reliable indicator of water stress—an abiotic limiting factor for photosynthesis. This factor can be explained by the fact that information from vegetation images is mainly interpreted by differences and changes in the green of plant leaves and the spectral characteristics of the canopy [64]. The plant water content has several primary and secondary effects on leaf characteristics, which in turn influence leaf, canopy and top-of-atmosphere spectral reflectance [10]. It is important to mention that, among the objectives of their work, Daughtry et al. [52] proposed a strategy to detect the state of leaf chlorophyll of plants using remote sensing data for corn (*Zea mays* L.) in the field. Among the results, the authors observed that spectral vegetation indices that combined reflectances of the near-infrared and other visible bands such as MCARI and NIR/Green were responsive to chlorophyll concentrations.

In this study, the NDVI, SAVI, OSAVI, MCARI and GDVI proved promising as monitoring tools of coffee. However, long-term studies for this perennial crop are needed, and other analysis methods are required to validate the indices and establish a relationship between them and coffee productivity. Under the experimental conditions, the SAVI, OSAVI, GDVI and MCARI proved efficient to indicate the need for soil amendment, as it improves the sensitivity of the NDVI [64], which is very responsive to background factors, e.g., the brightness and shadow of vegetation canopies and soil background brightness [58]. Likewise, canopy temperature and the TCARI can be used as indicators of water-stressed plants.

All vegetation indices, with the exception of the MTCI, were negatively correlated with canopy temperature. This finding could show that higher temperatures may be associated with lower chlorophyll content or stress in coffee trees. The indices NDVI, NDRE and MCARI, which are related to indices of leaf vigor, leaf density and chlorophyll content, were strongly correlated with each other. This correlation suggests that they are consistently good and aligned indicators of plant vigor. Barata et al. [39] found positive correlations (>0.7) between plant growth and the NDVI, MSAVI, NDRE, GNDVI and MCARI. This finding is consistent with our observations; although the variable growth was not specifically addressed, it is also related to tree vigor. Productivity was strongly and positively correlated with physiological variables (>0.80), but negatively with temperature (−0.62) in 2020 (Figure 8). This finding indicates that high chlorophyll levels and vegetative vigor (leaf density) are beneficial for productivity, while high plant canopy temperatures can be harmful.

Correlations between VIs and productivity suggest that remote sensing techniques using these indices can be effective in predicting coffee tree yield and detecting plant vigor problems at an early stage. This study contributes to reinforcing the possibilities of integrating remote sensing technologies and field data in modern agriculture for a more efficient and sustainable management of coffee. On the other hand, the strong correlations between vegetation indices with stomatal conductance and transpiration highlight the integration of these processes in the general performance of plants and the importance of water availability.

### 4.2. Grouping of Different Genotypes and Water Regimes

Principal component analyses detected considerable variation in the NDVI, OSAVI, GDVI and SAVI, which were the variables that explained the conditions within the water regime group and genotypes (Figure 9). Because these indices are strongly correlated and sensitive to changes in water availability, they can be used to explain the greater variation within groups of different water regimes [58].

The PCA showed overlapping of the treatments WD 100, WD 50 and rainfed water regimes (Figure 9), indicating that they are very similar systems for the indices TCARI and canopy temperature. Likewise, the correlations between T and physiological variables (gs, A and E), vegetation indices and plant productivity were negative (Figure 6) because drought-stressed plants close the leaf stomata to retain the water in the plant. This action decreases stomatal conductance, transpiration and increases leaf temperature [43,64,69,70,71]. On the other hand, when the stomata are open, water evaporates from the leaf through transpiration, which cools the leaf [64]. Therefore, crop canopy temperature has been accepted as an indicator of crop drought stress [70].

Regarding the grouping of water regimes, most of the variables studied were more closely aligned with the treatments that supplied more water (FI 100 and FI 50), especially in 2019 (Figure 9A). This finding indicates that a greater water supply resulted in optimal conditions for plants, as indicated by vegetation indices related to plant vigor. In contrast, the variables TCARI and T were correlated with lower plant performance in 2019. It is worth noting that the TCARI is very sensitive to chlorophyll content and strongly influenced by soil reflectance and plant canopy [58]. Knowing this information, this index should be interpreted parsimoniously, since the effects that alter its values are systemic and complex. In 2020, in addition to canopy temperature, the indices MTCI and NDRE were grouped in the rainfed treatment, indicating thermal or drought stress under these conditions (Figure 9C). The NDRE is commonly associated with pigment maturation, including chlorophyll (strongly linked to vegetative vigor) and is widely used in crop monitoring [39]. Bento et al. [37] also observed the sensitivity of this index to thermal elevation and precipitation in coffee. The water regimes FI 100 and FI 50 were grouped close together, especially in 2020, suggesting similar responses to irrigation in spite of the different water volumes (Figure 9A). The water stress treatments WD 100, WD 50 and the rainfed regime, where water was supplied by rainfall only, also formed close clusters in both study years, indicating noticeable effects of water reduction and a similar grouping of water stress regimes.

The different irrigation regimes induced distinct response patterns in the trees, as shown by the clusters (Figure 9A,C). Irrigation strategies, as in FI 100 and FI 50, induced similar responses, indicating that even a 50% reduction in water replacements does not significantly alter the plant response compared to the total water replacement of evapotranspiration. For Barbosa et al. [36], the mixture of pixels between branches and leaves in the crop canopy can affect the discrimination of indices. In other words, as the effects of the vegetation cover of the canopy in these two treatments were similar (Figure 4, Figure 5 and Figure 6), the results of the treatments were possibly not correctly captured by the VIs. Total water replacement, both continuous and seasonal, provided better conditions for growth and development, as shown by the formation of groups with vegetation indices, more aligned with productivity in 2019 and 2020. On the other hand, treatments with reduced water supply diverged from these indices, indicating a response to stress conditions in the coffee trees. These results are crucial for irrigation management in coffee, especially in drought-prone environments or where water is a limited resource.

In the analysis of genotype groups, the proximity and alignment of the vectors of physiological factors and most of the VIs indicated a positive relationship between high plant vigor and high metabolic activity (Figure 7 and Figure 9B). On the other hand, the TCARI vectors and plant canopy temperature diverged from the other vegetation indices, which suggests that higher temperatures are adversely related with plant vigor. In 2020, in addition to canopy temperature, the MTCI was also grouped more closely to the TCARI.

The principal component analysis showed no clear distinction between the different genotypes CATUAI 62, E 237 and IAPAR 59. This finding may suggest that the genotypic differences are not as marked as the effects of irrigation treatments on the variables evaluated in 2019 (Figure 9A,B). This variability within groups of genotypes may be influenced by the genetic characteristics of the plants, which would be an interesting aspect for cultivar recommendation.

## 5. Conclusions

In this study, coffee genotypes responded to variations in water availability, providing valuable information for genotype selection based on water use efficiency and tolerance to drought stress. Genotype Iapar is adapted to water stress, while Catuaí 62 and E237 are not adaptable and should not be grown in drought-prone areas or regions. This study, therefore, discriminated adequate genotypes for regions with limited water availability and tested irrigation regimes to maximize water use efficiency, considering the specific response of each coffee genotype. Another important aspect was the genetic variability in the genotypes chosen for cultivation, especially in the current context of climate change and decreasing water resources available for irrigation. The use of VIs to evaluate the temporal performance of the crop in response to different water managements demonstrated that the indices were efficient in discriminating the best genotypes, the optimal water conditions for each genotype and evaluating the performance of the plants, constituting an efficient tool for plant phenotyping for drought tolerance and water use efficiency of coffee genotypes. The use of VIs correlated with other climatic variables, such as temperature and relative humidity, at different times of the year, can be useful to estimate the productivity of coffee genotypes under different water managements.

## Figures and Tables

**Figure 1 sensors-24-07271-f001:**
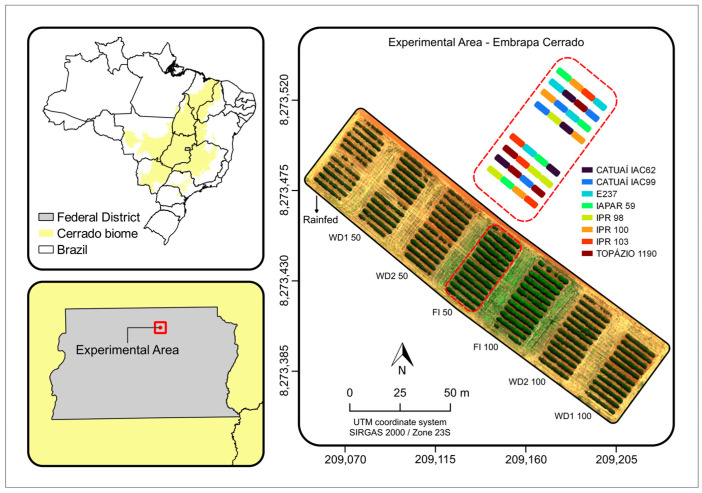
Location and arrangement of treatments and genotypes in the study area at Embrapa Cerrados in Brasília, DF, in the Cerrado biome.

**Figure 2 sensors-24-07271-f002:**
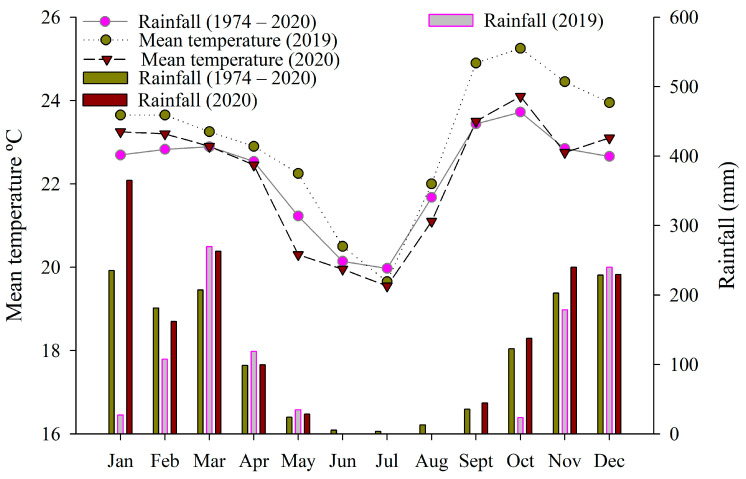
Long-term maximum and minimum average climate data over the past 46 years (1974–2020) in the area.

**Figure 3 sensors-24-07271-f003:**
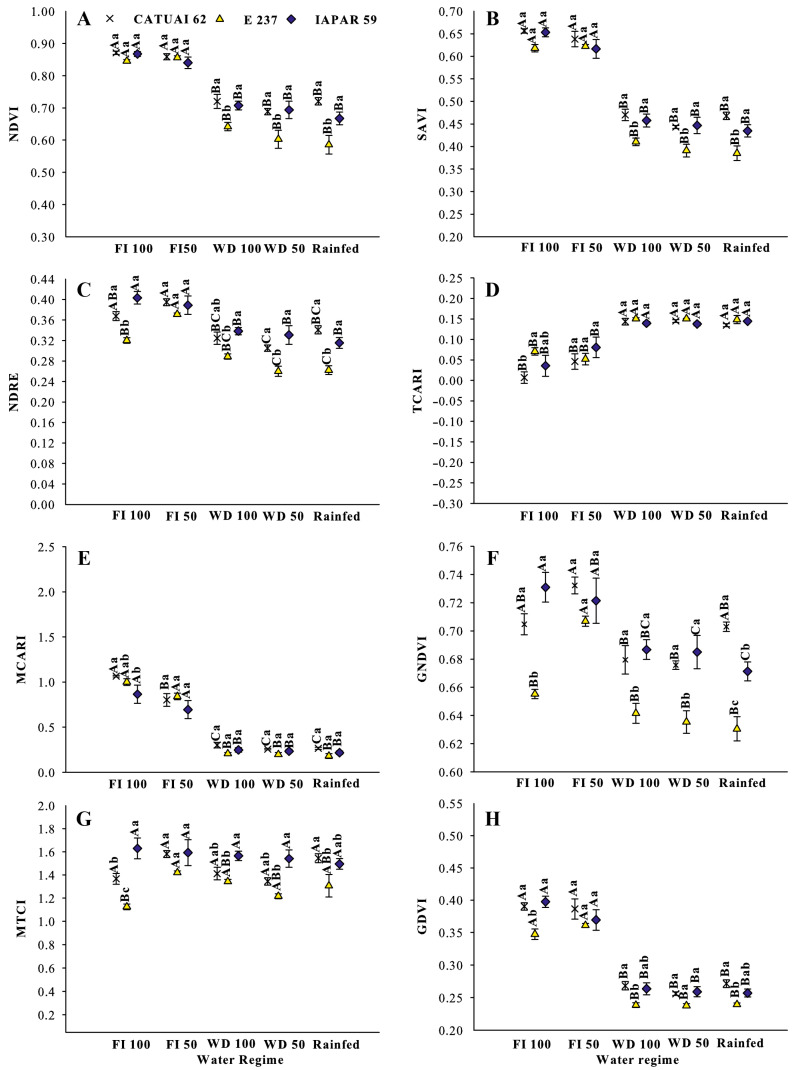
Vegetation indices (**A**) NDVI, (**B**) SAVI, (**C**) NDRE, (**D**) TCARI, (**E**) MCARI, (**F**) GNDVI, (**G**) MTCI and (**H**) GDVI, in 2019, for three coffee genotypes (Catuaí 62, E237 and Iapar 59) under five water regimes (FI 100%, FI 50%, WS 100%, WS 50% and Rainfed) during water stress in drought treatments in 2019. Means followed by the same capital letters compare water regimes for each coffee genotype, and lowercase letters compare coffee genotypes within each water regime.

**Figure 4 sensors-24-07271-f004:**
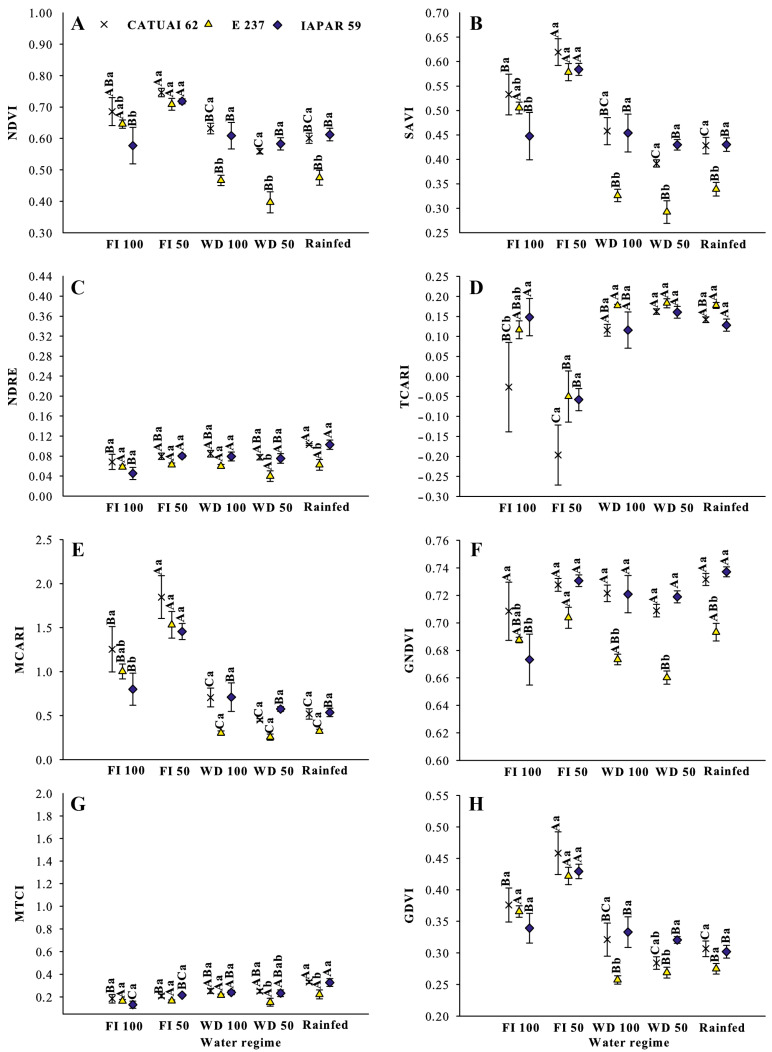
Vegetation indices (**A**) NDVI, (**B**) SAVI, (**C**) NDRE, (**D**) TCARI, (**E**) MCARI, (**F**) GNDVI, (**G**) MTCI and (**H**) GDVI, in 2020, for three coffee genotypes (Catuaí 62, E237 and Iapar 59) under five water regimes (FI 100%, FI 50%, WS 100%, WS 50% and Rainfed) during water stress in drought treatments in 2019. Means followed by the same capital letters compare water regimes for each coffee genotype, and lowercase letters compare coffee genotypes within each water regime.

**Figure 5 sensors-24-07271-f005:**
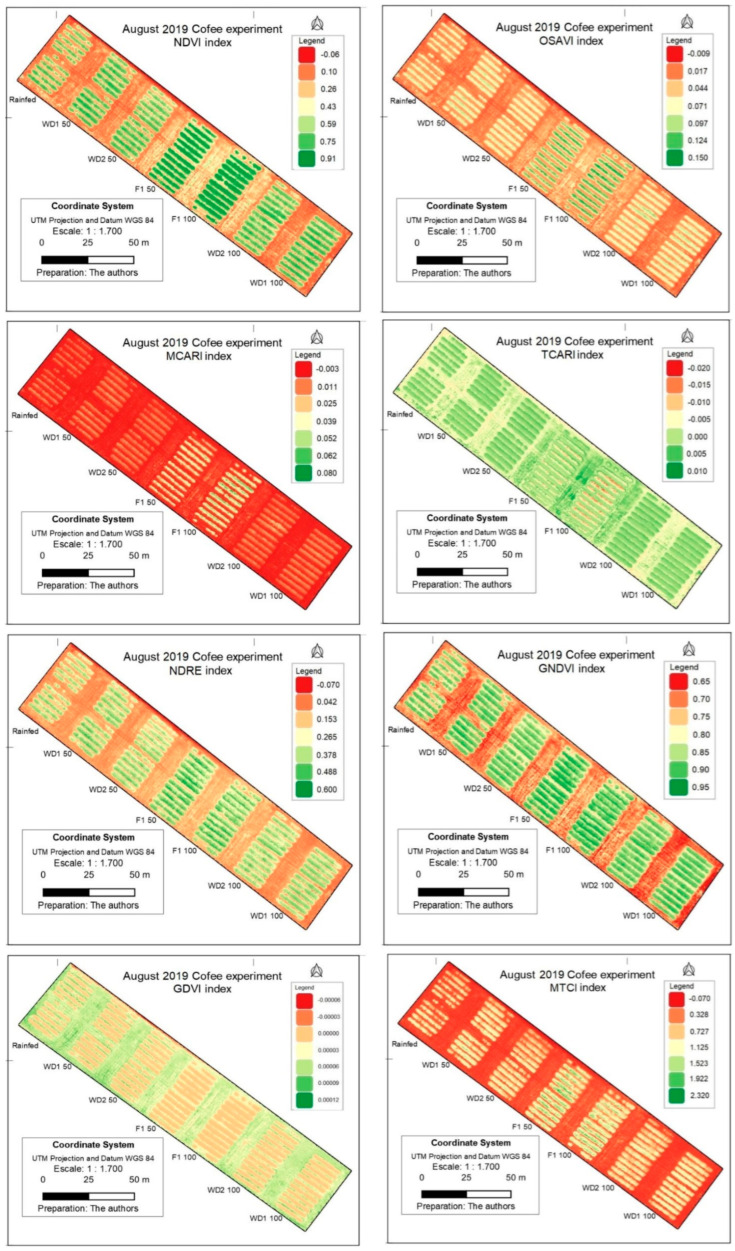
Vegetation indices NDVI, OSAVI, MCARI, TCARI, NDRE, GNDVI, GDVI and MTCI evaluated for three coffee genotypes (Catuaí 62, E237 and Iapar 59) in response to five water regimes (from left to right, rainfed; WD (water stress) 1, 50%; WD2 50%; FI (full irrigation) in 2019.

**Figure 6 sensors-24-07271-f006:**
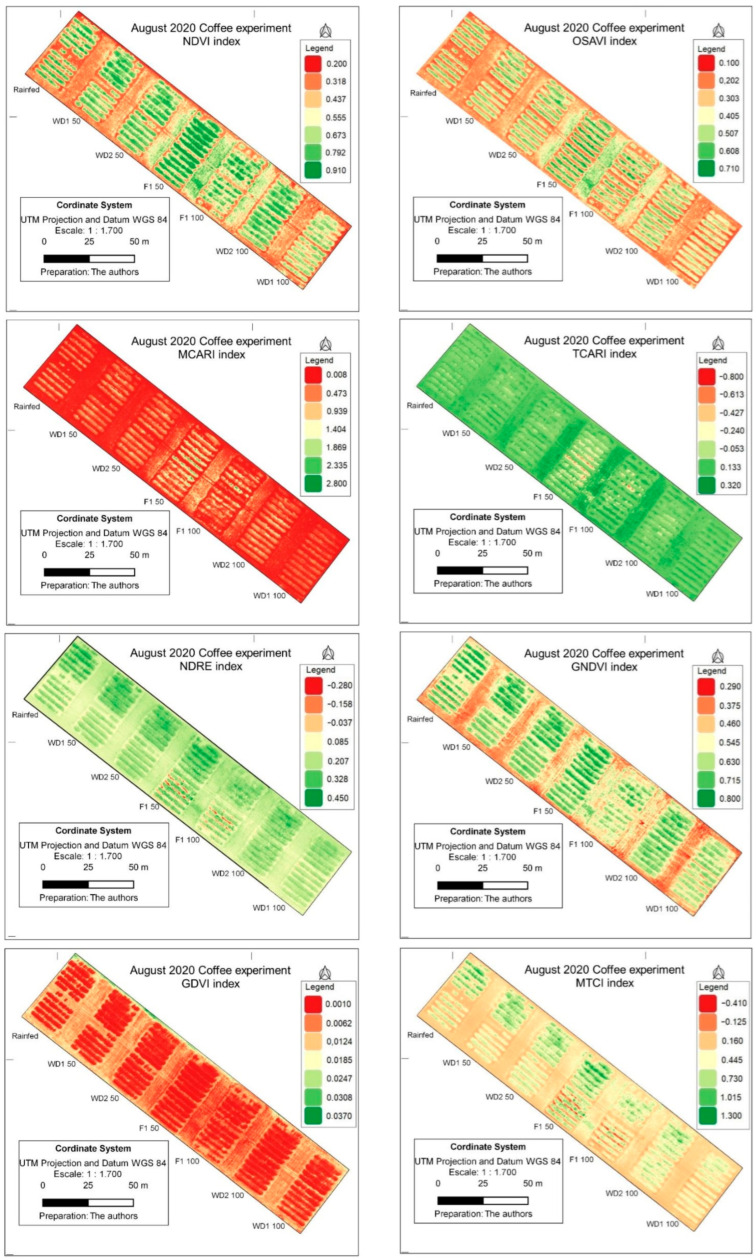
Vegetation indices NDVI, OSAVI, MCARI, TCARI, NDRE, GNDVI, GDVI and MTCI evaluated for three coffee genotypes (Catuaí 62, E237 and Iapar 59) in response to five water regimes (from left to right, rainfed; WD (water stress) 1, 50%; WD2 50%; FI (full irrigation) in 2020.

**Figure 7 sensors-24-07271-f007:**
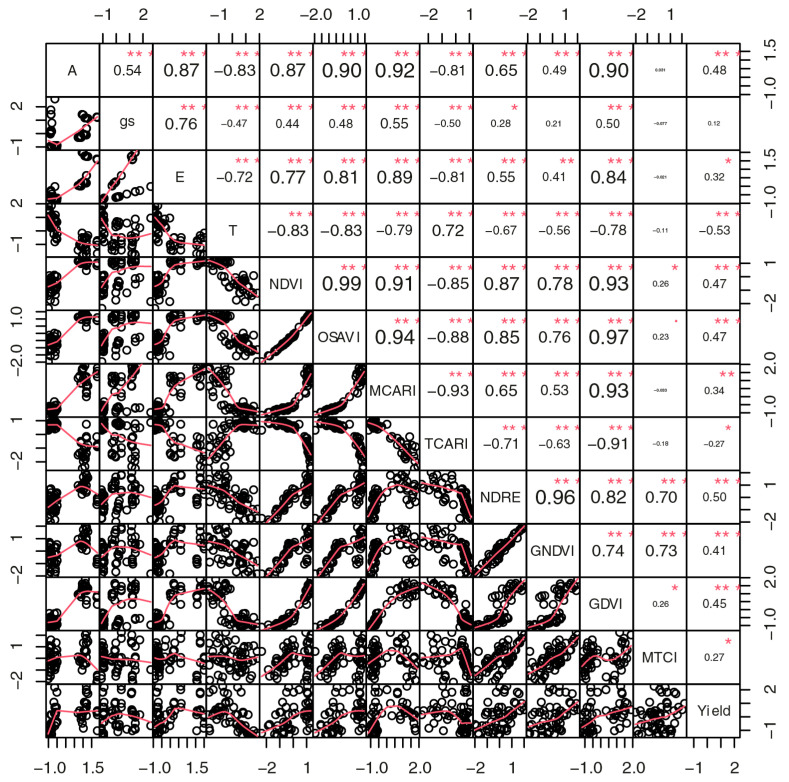
Correlogram of Pearson’s correlation between vegetation indices and leaf gas exchange in 2019. A: photosynthesis, gs: stomatal conductance, E: transpiration and T: canopy temperature °C. Values close to 1 indicate a strong positive correlation, while values close to −1 represent a strong negative correlation. Values close to 0 suggest an absence of significant linear relationship. In addition, the size of the source observed in the corelogram was used to graphically represent the magnitude of the correlation, allowing a clear visualization of the strength of the associations. Stronger correlations (positive or negative) were highlighted with larger font size and bold, while weaker correlations were presented with smaller and unbold fonts. The statistical significance of the correlations was evaluated at a probability level: * significant at *p* < 0.05, ** significant at *p* < 0.01, *** significant at *p* < 0.001.

**Figure 8 sensors-24-07271-f008:**
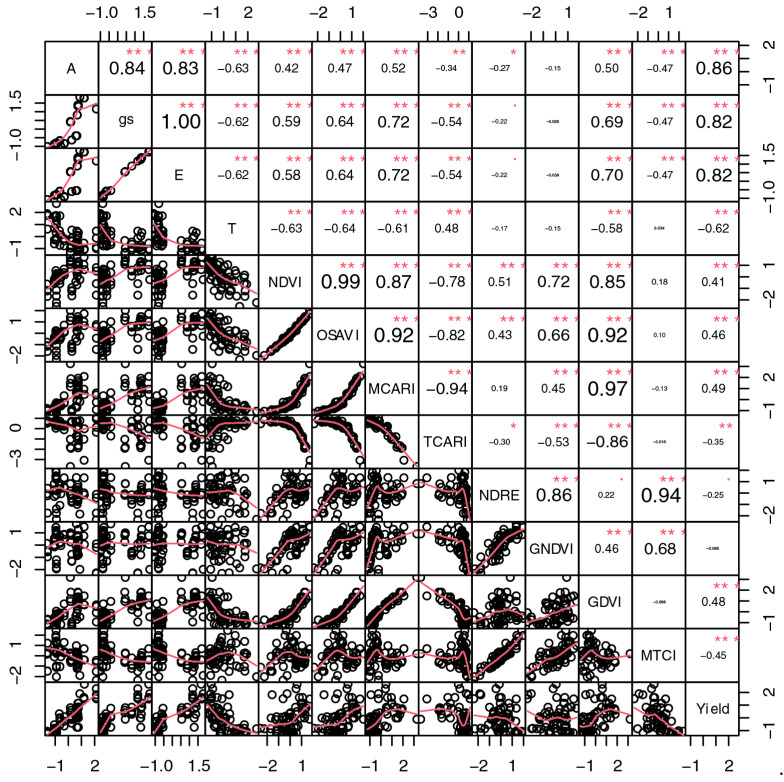
Correlogram of Pearson’s correlation between vegetation indices and leaf gas exchange in 2020. A: photosynthesis, gs: stomatal conductance, E: transpiration and T: canopy temperature °C. Values close to 1 indicate a strong positive correlation, while values close to −1 represent a strong negative correlation. Values close to 0 suggest an absence of significant linear relationship. In addition, the size of the source observed in the correlogram was used to graphically represent the magnitude of the correlation, allowing a clear visualization of the strength of the associations. Stronger correlations (positive or negative) were highlighted with larger font size and bold, while weaker correlations were presented with smaller and unbold fonts. The statistical significance of the correlations was evaluated at a probability level: * significant at *p* < 0.05, ** significant at *p* < 0.01, *** significant at *p* < 0.001.

**Figure 9 sensors-24-07271-f009:**
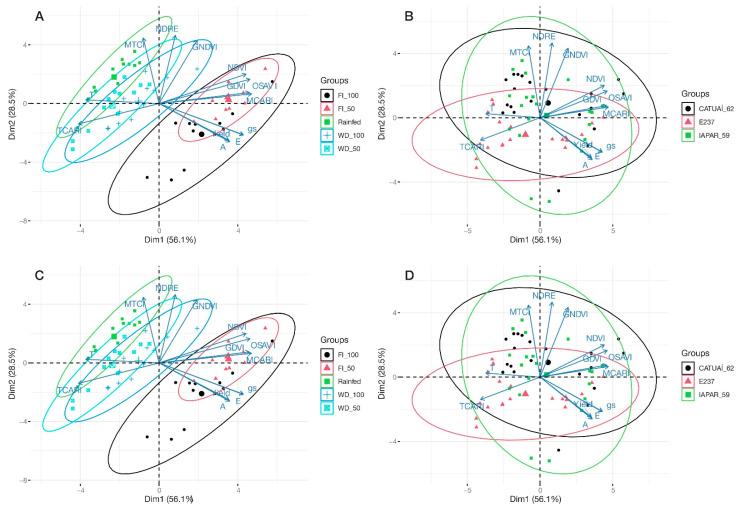
Exploratory analysis of the principal components for vegetation indices and coffee tree physiology and productivity in response to different water regimes and of different genotypes in the growing seasons of 2019 (**A**,**B**) and 2020 (**C**,**D**). The vectors of the variables projected onto the graphs indicate the magnitude and direction of their contribution to the separation between groups. The length of the vectors reflects the intensity of their influence on the principal components, while their orientation highlights the multivariate differences between the evaluated conditions. The ellipses drawn around the experimental groups represent the intragroup dispersion based on the covariance of the data. The center of each ellipse corresponds to the centroid of the respective group, representing the average position of the observations. The orientation and size of the ellipses indicate the internal variability of each group: more compact ellipses suggest greater homogeneity within the observations. In contrast, larger ellipses indicate greater heterogeneity, possibly associated with the different conditions imposed by the treatments. The distribution of the vectors and the separation of the ellipses demonstrate that the analyzed variables play a significant role in differentiating between the groups of cultivars and irrigation regimes, providing insights into the physiological and spectral responses under each evaluated scenario.

**Table 1 sensors-24-07271-t001:** Micasense RedEdge-MX^®^ multispectral camera, technical specifications.

MicaSense Altum^®^ Camera	
Manufacturer	Ag Eagle (Micasense), North Wichita, KS, USA
Weight	231.9 g
Size	8.3 cm × 5.9 cm × 4.54 cm
Spectral band Wavelength (nm)	Blue (475 nm wavelength, 32 nm bandwidth)
	Green (560 nm wavelength, 27 nm bandwidth)
	Red (668 nm wavelength, 16 nm bandwidth)
	Rededge (717 nm wavelength, 10 nm bandwidth)
	NIR (840 nm wavelength, 40 nm bandwidth)
Sensor Resolution	1.228 MP per band (1280 × 960 pixels)
Field of View (FOV)	47.2 degrees horizontal; 35.4 degrees vertical

**Table 2 sensors-24-07271-t002:** FLIR Duo Pro^®^ thermal camera, technical specifications.

Camera FLIR Duo Pro^®^	
Weight	325 g
Size	85 mm × 81.3 mm × 68.5 mm (3.35 in) × 3.20 in × 2.70 in
Spectral band	7.5–13.5 µm
Sensitivity	<50 mK
Sensor Resolution	336 × 256 pixels
Field of View (FOV)	56° × 45°

**Table 3 sensors-24-07271-t003:** Parrot Sequoia^®^ multispectral camera, technical specifications.

Camera Parrot Sequoia^®^	
Manufacturer	Ag Eagle (Micasense), North Wichita, KS, USA
Weight	72 g
Size	2.9 cm × 5.9 cm × 4.1 cm.
Spectral band Wavelength (nm)	Green (550 nm wavelength, 40 nm bandwidth)
	Red (660 nm wavelength, 40 nm bandwidth)
	Rededge (735 nm wavelength, 10 nm bandwidth)
	Near Infrared (790 nm wavelength, 40 nm bandwidth)
Sensor Resolution	1.228 MP per band (1280 × 960 pixels)
Field of View (FOV)	70° HFOV

**Table 4 sensors-24-07271-t004:** Vegetation indices calculated from multispectral images.

Indices	Equation	References
NDVI	$\frac{\left(NIR-R\right)}{\left(NIR+R\right)}$	Rouse et al. [28]
OSAVI	$\frac{\left(NIR-R\right)}{\left(NIR+R+0.16\right)}$	Rondeaux et al. [51]
MCARI	$\left[\left(RE-R\right)-0.2\times \left(RE-G\right)\right]\times \left(\frac{RE}{R}\right)$	Daughtry et al. [52]
TCARI	$3\times \left[\left(RE-R\right)-0.2\times \left(RE-G\right)\times \left(\frac{RE}{R}\right)\right]$	Haboudane et al. [33]
NDRE	$\frac{\left(NIR-RE\right)}{\left(NIR+RE\right)}$	Gitelson and Merzlyak [53]
GNDVI	$\frac{\left(NIR-G\right)}{\left(NIR+G\right)}$	Gitelson et al. [54]
GDVI	$\frac{\left({NIR}^{2}-R^{2}\right)}{\left({NIR}^{2}+R^{2}\right)}$	Wu [34]
MTCI	$\frac{\left(NIR-RE\right)}{\left(RE+R\right)}$	Dash and Curran [35]

NDVI: normalized difference in red edge; OSAVI: optimized soil adjusted vegetation index; MCARI: modified chlorophyll and reflectance index; TCARI: transformed chlorophyll absorption ratio; NDRE: normalized difference in red edge; GNDVI: green normalized difference vegetation index; GDVI: generalized difference vegetation index; MTCI: MERIS terrestrial chlorophyll index (MERIS: resolution imaging spectrometer); RE: red edge channel; G: green channel; NIR: reflectances in the near-infrared; R: red channels.

## Data Availability

All coffee genotypes used in this manuscript are available on the market and registered by the Ministry of Agriculture: https://sistemas.agricultura.gov.br/snpc/cultivarweb/cultivares_registradas.php (accessed on 2 September 2024).

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
