# Peer review of "Multispectral Images for Drought Stress Evaluation of Arabica Coffee Genotypes Under Different Irrigation Regimes"

_sensors, 2024, doi:10.3390/s24227271_

Round 1
Reviewer 1 Report
Comments and Suggestions for Authors
The authors evaluated the impact of growing different cultivars and irrigation strategies on coffee productivity in a significant coffee-producing and exporting country worldwide. Given the high value of the cash crop and its susceptibility to extreme weather events, this study attempts to address important science questions.
The manuscript could be improved further if the following pointers are addressed:
1) In the abstract (on line 35), it might be more informative if you specify "dry season" next to "June to September." This might help readers understand the design of experiments better.
2) In the abstract (line 44), readers may not immediately understand TCARI. It might be a good idea to say "temperature-based indices, such as canopy temperature and TCARI,...."
3) The abstract could perhaps end with a statement explaining why this is a significant problem to address, given the value of the cash crop and its susceptibility to droughts.
4) The first paragraph in the Introduction could provide more information about the Coffee crop grown in Brazil, such as the typical sowing and harvest dates and whether it is an annual, biennial, or perennial crop.
5) Line 58 in the first paragraph of the Introduction -- "However, extreme weather events, e.g., prolonged droughts and frosts, have affected the production." Please add citations at the end of this sentence.
6) Line 71 in the second paragraph of the Introduction -- "The spatial, spectral temporal and radiometric image resolution of the sensors is high....". Please be more specific about what sensors are being referred to here.
7) Line 76 in the second paragraph of the Introduction -- "There are also vegetation indices that correlate well with the biophysical vegetation parameters and are widely used to estimate biomass and changes in crop growth and development." Please add citations at the end of the sentence.
8) As per the manuscript, 2020 seems to be a drought year compared to 2019. It's interesting to see how the time series of mean temperature in 2020 is below the 2019 plot in Figure 2.
9) Section 2.5 could benefit from a better explanation of the approach. Perhaps the section could start with a summary of what is being done. For example, "We would like to study the impact of water regimes and genotypes on each vegetation index." Also, in line 194, "ANOVA was performed in randomized blocks for each variable for each water regime." Please clarify what "variable" refers to.
10) The data presented in Figures 3 and 4 look the same. Please check if the data used to plot these Figures is correct.
11) Please consider alternative ways to visualize the data in Figures 3 and 4. Could they be combined into one Figure to enable easier comparison between 2019 and 2020? Right now, the line plots in Figures 3 and 4 for the three genotypes mostly overlap, making it difficult to read. The y-axis range in all the subplots could be constrained further to make it easier to read. Another way could be to use a bar plot instead of a line plot.
12) Figure 3 caption (lines 232-233) is not very clear - "Means followed by the same capital letters compare water regimes within each genotype and lowercase letters compare genotypes within each water regime".
13) Please add the location of water regimes (FI 100%, FI 50%, WS 100%, WS 50% and Rainfed) in each subplot in Figures 5 and 6. For example, in Figure 5, the MCARI index seems high for FI 50% and FI 100% plots. Adding the water regime labels would make it easier to comprehend.
14) Given that 2020 was a drought year, it's interesting to see how the NDVI and OSAVI maps look brighter (with higher values) in 2020 compared to 2019 (Figures 5 and 6). The higher temperatures in 2020 should have resulted in lower values of these indices.
15) It's important to note that Pearson correlation measures the strength of the linear relationship between variables and not the overall relationship (linear and nonlinear). Vegetation productivity has been shown to vary nonlinearly with some vegetation and climate indices. This aspect could be highlighted in the discussion section.
16) Section 3.3: There are a few typos in references to Figures. For example, line 362 should be Figure 9B and 9D (and not 7D). Similarly, line 369 should be Figure 9A and 9C (and not 7C).
17) Please consider moving the first two paragraphs of the discussion section to the Introduction.
Comments on the Quality of English LanguageAs the pointwise comments listed above highlight, some of the sentences and sections could be rewritten to improve clarity.
Reviewer 2 Report
Comments and Suggestions for Authors
In summary, the manuscript entitled "Multispectral images for drought stress evaluation of Arabica coffee genotypes under different irrigation regimes" presents a comprehensive study on the use of remote sensing technology to assess drought stress in coffee plants. The authors have successfully demonstrated the potential of vegetation indices (VIs) in monitoring the physiological status of coffee genotypes under varying water availability conditions. The study is methodologically sound, with a well-designed experiment, proper data analysis, and a clear presentation of results. The findings are relevant to the field of precision agriculture and contribute to the understanding of how different irrigation regimes affect coffee plant performance.Nevertheless, there are certain sections and sentences that require clarification and some questions that demand answers.
1. Introduction
1. The section could be supplemented with information on the latest research advancements in the use of unmanned aerial vehicles (UAVs) and spectral analysis in coffee bean research.
1. Materials and Methods
2.1 Study area and experimental design
1. Why were Three genotypes (Catuaí 62, E237, and Iapar 59) selected, given that the initial setup mentioned eight genotypes?
2. More background and rationale for choosing these three coffee genotypes for the study should be provided.
2.2 Sensors and image acquisition
1. Did the different spectral lenses used in the two flights affect the spectral data obtained?
2. Table 2: "Weight 72 g." should be corrected to "Weight: 72 g".
2. Results
1. The data in Figures 3 and 4 are almost identical; please attach the data.
2. Figure 9: The authors should ensure that all figures and images have sufficient clarity and resolution to be clearly displayed in electronic format.
3. Discussion
1. Expanding on the rationale for selecting vegetation indices (VIs) and their relevance to drought stress detection would improve readability.
2. Clarifying the role of each index in drought response and their relationship with physiological characteristics (such as NDVI, OSAVI, GDVI) can strengthen the connection between data collection and the conclusions drawn.
4. Conclusions
1. The authors are encouraged to include an economic analysis in the discussion section, assessing the impact of different irrigation regimes and genotype selection on the economic benefits of coffee cultivation.
References
1. Reference 5: Formatting errors (repeated years).
Reviewer 3 Report
Comments and Suggestions for Authors
The manuscript, “Multispectral Images for Drought Stress Evaluation of Arabica Coffee Genotypes Under Different Irrigation Regimes,” explores the use of vegetation indices (VIs) derived from UAV-based multispectral imaging to evaluate the drought stress and performance of coffee genotypes. While the topic is relevant, the manuscript falls short in several areas that significantly limit its contribution to the field. The current form is not acceptable for publication, and major revisions are required, particularly in the methodological details, clarity of results interpretation, and consistency of the statistical analysis.
1. Line 51-92: The introduction does not sufficiently cover the limitations of previous studies in the use of Vis for coffee drought stress evaluation. There is a need for a more thorough literature review to better position the current study in the context of existing research.
2. Line 98: What is “Aw”?
3. Line 111: What is “0.74-ha”?
4. Line 127-128: Would it be possible for you to make those data available for reproducibility?
5. Table 3: Please provide the full names of those indices when they appear for the first time in the manuscript and explain the notations in those equation such as what NIR is, what R is … ?
6. For MCARI and TCARI, what do you mean by “0,2”?
7. Line 191-200: The statistical analysis section may not be adequate since it is important. More details of the methods should be provided. You only mention ANOVA but do not provide details on any assumptions checking, post-hoc analysis, or normalization. This kind of omission may raise concerns about the validity of the results.
8. Line 205: When any abbreviation first appears (such as Vis), you should provide its full name.
9. Figure 3 and 4 are in low resolution and these two figures are difficult to read. The graphs contain busy information, making it challenging to discern the key trends. It would be helpful to simplify the figures or use separate panels for clarity.
10. Line 232-233, 238-239: The meaning of this sentence (“Means followed by …”)is unclear.
11. The quality of Figure 9 is poor.
12. Line 381-384: The introduction of vegetation indices should be in the introduction section.
13. Line 392-395: Please rephrase this running sentence. It is not clear.
14. Line 435-500: Those background information about those indices should be provided in earlier section.
15. There are several instances of grammatical errors and awkward phrasing throughout the manuscript, which detract from the clarity of the writing. The manuscript would benefit from a thorough language review.
Comments on the Quality of English Language
There are several instances of grammatical errors and awkward phrasing throughout the manuscript, which detract from the clarity of the writing. The manuscript would benefit from a thorough language review.
Round 2
Reviewer 3 Report
Comments and Suggestions for Authors
The revision is okay. I don't have more further comments.
Author Response
The third reviewer commented that "The revision is okay. I don't have further comments".